# Traffic Agents Trajectory Prediction Based on Spatial–Temporal Interaction Attention

**DOI:** 10.3390/s23187830

**Published:** 2023-09-12

**Authors:** Jincan Xie, Shuang Li, Chunsheng Liu

**Affiliations:** 1School of Information and Automation Engineering, Qilu University of Technology (Shandong Academy of Sciences), Jinan 250353, China; 202014800@mail.sdu.edu.cn (J.X.); liuchunsheng@sdu.edu.cn (C.L.); 2School of Control Science and Engineering, Shandong University, Jinan 250061, China

**Keywords:** trajectory prediction, spatial–temporal interaction, social interaction

## Abstract

Trajectory prediction aims to predict the movement intention of traffic participants in the future based on the historical observation trajectories. For traffic scenarios, pedestrians, vehicles and other traffic participants have social interaction of surrounding traffic participants in both time and spatial dimensions. Most previous studies only use pooling methods to simulate the interaction process between participants and cannot fully capture the spatio-temporal dependence, possibly accumulating errors with the increase in prediction time. To overcome these problems, we propose the Spatial–Temporal Interaction Attention-based Trajectory Prediction Network (STIA-TPNet), which can effectively model the spatial–temporal interaction information. Based on trajectory feature extraction, the novel Spatial–Temporal Interaction Attention Module (STIA Module) is proposed to extract the interaction relationships between traffic participants, including temporal interaction attention, spatial interaction attention, and spatio-temporal attention fusion. By adaptive allocation of attention weights, temporal interaction attention is a temporal attention mechanism used to capture the movement pattern of each traffic participant in the scene, which can learn the importance of historical trajectories at different moments to future behaviors. Since the participants number in recent traffic scenes dynamically changes, the spatial interaction attention is designed to abstract the traffic participants in the scene into graph nodes, and abstract the social interaction between participants into graph edges. Coupling the temporal and spatial interaction attentions can adaptively model the temporal–spatial information and achieve accurate trajectory prediction. By performing experiments on the INTERACTION dataset and the UTP (Unmanned Aerial Vehicle-based Trajectory Prediction) dataset, the experimental results show that the proposed method significantly improves the accuracy of trajectory prediction and outperforms the representative methods in comparison.

## 1. Introduction

Trajectory prediction aims to predict the movement intention of traffic participants in the future based on the historical observation trajectories, which plays an important role in intelligent transportation, self-driving, intelligent robots, etc. Compared with traditional trajectory prediction for just pedestrians, urban traffic scenarios involve pedestrians, vehicles and other traffic participants, which bring more difficulties. In this study, we focus on the traffic participants trajectory prediction task.

The trajectory prediction methods can be divided into traditional methods and deep-learning-based methods.

Traditional methods use manually designed models to predict trajectory, including the model of social forces [1], behavioral model [2], nonparametric statistical model [3], hidden Markov model [4], etc. For complex and crowded environments, the generalization of these methods is usually poor.

Compared with traditional trajectory prediction methods, the advantage of deep-learning-based trajectory prediction methods is that they do not rely on artificial design features, and can learn complex and diverse social interactions through training. The mainstream deep learning trajectory prediction methods are divided into Recurrent Neural Networks (RNN) [5,6,7], Generative Adversarial Networks (GAN) [8,9,10] and Graph Neural Networks (GNN) [11,12,13]. The RNN-based methods cannot achieve parallel processing, and the training speed and prediction performance usually need further improvement when dealing with multi-modal trajectories. The GAN-based methods can generate a trajectory distribution with multiple modalities, and also have some problems including unstable training and pattern collapse. The GNN-based methods can intuitively construct the topological relationship between traffic participants, and usually suffer degradation problems with deepening layers.

Based on these analyses, though all these methods based on RNN, GAN and GNN can model trajectories, the attention mechanisms [14,15,16,17] are key to effectively model the social interaction of surrounding traffic participants in both time and spatial dimensions. Furthermore, compared with the trajectory prediction task for just pedestrians, traffic participants including pedestrians, vehicles and others are with more complex behavioral patterns. Their motion states are affected by the historical trajectory and the social interaction of surrounding traffic participants. It is crucial to consider both the spatial and temporal interaction information. Yet, these previous methods just use pooling methods to simulate the interaction process between traffic participants, and cannot fully capture the spatio-temporal dependence between traffic participants in complex urban traffic scenarios, which may lead to the generation of some unreasonable prediction trajectories.

To overcome these problems, we propose the Spatial–Temporal Interaction Attention-based Trajectory Prediction Network (STIA-TPNet), which can effectively model the spatial–temporal interaction information. The trajectory prediction task is treated as a sequence prediction problem. The trajectory feature extraction module takes the historical trajectory information as input, and completes the extraction of key information by using embedding layers and LSTM networks. Based on trajectory feature extraction, the novel Spatial–Temporal Interaction Attention Module (STIA Module) is proposed to extract the interaction relationships between traffic participants, including temporal interaction attention, spatial interaction attention, and spatio-temporal attention fusion. The temporal interaction attention is a temporal attention mechanism used to capture the movement pattern of each traffic participant by adaptive allocating attention weights, which can learn the importance of historical trajectories at different moments to the future behaviors. Based on GNN structure to represent the dynamically changing participants, the spatial interaction attention is designed to abstract the participants in the scene into graph nodes, and abstract the social interaction between participants into graph edges. Coupling the temporal and spatial interaction attentions can adaptively model the temporal–spatial information and achieve accurate trajectory prediction. By performing experiments on the INTERACTION dataset [18] and the UTP (Unmanned Aerial Vehicle-based Trajectory Prediction) dataset, the experimental results show that the proposed method significantly improves the accuracy of trajectory prediction and outperforms the representative methods in comparison.

The remainder of this paper is organized as follows. We give a brief introduction to review the related work in Section 2. Section 3 describes the proposed methods. The extensive experiments are presented in Section 4, and Section 5 gives the final conclusions.

## 2. Related Works

In the trajectory prediction task for traffic scenarios, there are difficulties, including multiple interactive participants, multi-modal driver behavior, and inherent uncertainty in driver motion. Yet, previous methods just focus on pedestrians. The trajectory prediction methods can be divided into prediction methods based on traditional methods and deep-learning-based methods. We review the trajectory prediction methods in this section.

### 2.1. Trajectory Prediction Based on Traditional Methods

The model of social forces is a method that quantifies the interactions between participants as attraction and repulsion [1]. The social force model can be improved to improve trajectory prediction for large populations [19]. The explicit energy functions can also be used to estimate the hidden properties of participants [2]. Although the social force model can simulate the motion, the model only considers the action of force. The Gaussian process is used to predict the trajectories of each participant and the future trajectories based on the interaction potential function [3].

The hidden Markov model is commonly used to predict trajectories, which usually include three stages of Gaussian hybrid modeling, trajectory clustering and hidden Markov probability coding [4]. The nonparametric hidden Markov model can be used to predict the future trajectory [20]. Kitani et al. used inverse optimal control to train and optimize the model. Hidden Markov Decision Process Model [21] uses inverse optimal control to train and optimize the model, and can more effectively learn the motion intent.

In conclusion, the traditional trajectory prediction methods are usually simple. For complex and crowded environments, the generalization ability is usually poor. In addition, hand-designed features and rules limit their ability to learn unpredictable features in the real scenarios.

### 2.2. Trajectory Prediction Based on Deep Learning

In recent years, deep learning technology has developed rapidly. Compared with traditional trajectory prediction methods, the advantage of deep learning-based trajectory prediction methods is that they do not rely on artificial design features, and can learn complex and diverse social interactions through training. The mainstream deep learning trajectory prediction methods are divided into Recurrent Neural Networks (RNN)-based methods, Generative Adversarial Networks (GAN)-based methods and Graph Neural Networks (GNN)-based methods.

**(1)** 
**Trajectory prediction methods based on Recurrent Neural Network.**


Introducing social functions to LSTM (Long Short-Term Memory), the Social LSTM [5] is proposed to treat the trajectory prediction problem as a sequence generation task. Based on the Social LSTM, a state refinement module [6] is proposed to adaptively refine the state of each traffic participant through a messaging framework. LSTM can also be improved with more LSTM models, to model the dependent information of participants, social rules, and scene environment [7].

Social Attention mechanism [14] can be used to describe the task of trajectory prediction; this model introduces social attention modeling of the influence of surrounding participants on the intention of target participants, overcoming the previous shortcomings of describing social interactions based on relative positions. Generating dense candidate targets based on probability [15] is an attention mechanism to select a trajectory. Zhao et al. [16] proposed a new multi-agent tensor fusion network with two parallel coded streams. Y-Net trajectory prediction network [17] used scene semantics and historical motion information to estimate the probability distribution and future position distribution of the final position. Using LSTM as social relation encoder, a Social Relation Attention-based Interaction aware LSTM (SRAI−LSTM) [22] utilizes social behavior to realize pedestrian trajectory prediction. This method continuously learns and updates the social relation feature from pedestrians’ relative positions. These methods reply on probabilities and attentions to generate the most possible trajectory.

As a popular structure for trajectory prediction, RNN-based methods have some problems. The training process is completed by minimizing the distance between the future trajectory and the groundtruth trajectory, so the model is biased towards predicting the average trajectory. However, the trajectory is multi-modal, so the prediction effect needs to be improved.

**(2)** 
**Trajectory prediction methods based on Generative Adversarial Network.**


Generative Adversarial Network (GAN) is a common method to generate trajectories. The Social Gnerative Adversarial Network (Social GAN) [8] integrates LSTM and GAN to perform trajectory prediction. Based on Social GAN, physical attention and social attention can be used to consider physical and social interactions [9]. The graph-based generative adversarial network [10] can be used to better mimic the social interaction of moving targets through a graph attention mechanism. Amirian et al. [23] proposed the Info-GAN structure, which effectively solved the problem of pattern collapse and decline of GAN. Dendorfer et al. [24] proposed a multi-generator trajectory prediction model MG-GAN, which significantly reduces the number of samples outside the distribution. The spatio-temporal graph attention (GSTA) network [25] combines GAN and LSTM to design spatial attention structure and feature updating mechanisms in spatial domain, and temporal attention and feature selection modules in the temporal domain. Experimental results show that the GSTA improves the pedestrian trajectory prediction accuracy. Based on GAN, a mixed Conditional Auto Encoder Generative Adversarial Network (CAE-GAN) [26] model using the multi-loss function is proposed. The GAN in the CAE-GAN is used to extract features of the generated trajectories and generate trajectories that are close to the real trajectories.

The GAN-based trajectory prediction method solves the problem of predicting only one “average good” trajectory, and can generate a trajectory distribution with multiple modalities. However, models based on GAN networks also have some problems, such as unstable training and pattern collapse.

**(3)** 
**Trajectory prediction methods based on Graph Neural Network.**


Graph Neural Network (GNN) is a common structure to perform trajectory prediction. Introducing a social spatio-temporal model to GNN can extract features by performing spatio-temporal convolution operations on the spatio-temporal graph [11]. Ivanovic et al. [12] proposed a multi-agent trajectory prediction framework Trajectron, which constructs participants into a dynamic temporal–spatial map to generate a multi-modal distribution of the future trajectories. The Trajectron model has been improved to the Trajectron++ version [13], which can generate dynamically feasible predictive trajectories for multiple traffic participants of different semantic types.

Constructing a graph to represent the interaction between adjacent participants can improve the performance [27]. The multi-scale hypergraph neural network GroupNet [28] can capture pairwise and grouped interactions, effectively modeling complex social impacts. Zhang et al. [29] combined Graph Attention Networks and convolutional gated recursive units to infer the future location of multiple traffic participants simultaneously. Mo et al. [30] proposed a heterogeneous edge-enhanced graph attention network to comprehensively simulate the interaction between participants. Based on Graph Convolutional Network and gated linear units, the STHGLU method [31] includes an adaptive model to understand the relationships between the participants, and includes a lightweight trajectory prediction model to trajectory prediction. In order to model the spatial and temporal relationship of multiple participants under uncertainties, the B-STAR model [32] combines spatio-temporal graph network and Bayesian in the transformer encoder and decoder architecture.

Compared with RNN- and GAN-based methods, the trajectory prediction method based on GNN can intuitively construct the topological relationship between participants, and the prediction performance is also improved. GNN has advantages with shallow networks; as the number of network layers deepens, the robustness of the network may decline.

Based on these analyses, though all these methods based on RNN, GAN and GNN can model trajectories, the attention mechanisms [14,15,16,17] are key to effectively model the social interaction of surrounding traffic participants in both time and spatial dimensions. Most these previous methods just use pooling methods to simulate the interaction process between participants and cannot fully capture the spatio-temporal dependence, and may accumulate errors with the increase in prediction time. To overcome these problems, we propose the Spatial–Temporal Interaction Attention-based Trajectory Prediction Network (STIA-TPNet), which effectively models the spatial–temporal interaction information with a temporal flow branch and a spatial flow branch.

## 3. Method

The Spatial–Temporal Interaction Attention-based Trajectory Prediction Network (STIA-TPNet) is proposed to effectively model the spatial–temporal interaction information. Based on trajectory feature extraction, the novel Spatial–Temporal Interaction Attention Module (STIA Module) is proposed to extract the interaction relationships between traffic participants, including temporal interaction attention, spatial interaction attention, and spatio-temporal attention fusion. The temporal interaction attention is a temporal attention mechanism used to capture the movement pattern of each participant in the scene. The spatial interaction attention is designed to abstract the participants in the scene into graph nodes, and abstract the social interaction between participants into graph edges. Coupling the temporal and spatial interaction attentions can adaptively model the temporal–spatial information and achieve accurate trajectory prediction. The details of the proposed methods are as follows.

### 3.1. Trajectory Prediction Definition

The trajectory prediction task is treated as a sequence prediction problem. Based on the historical trajectories of the moving targets, social interaction information and the surrounding environment are modeled to understand the movement intention of the target; and the future trajectories of all participants can be estimated. The definition is as follows.

There are *N* interaction traffic participants $A_{1},A_{2},\ldots ,A_{N}$. Each participant $A_{i}$ corresponds to a semantic category $C^{i}$; $C^{i}=0$ means $A_{i}$ is a vehicle; $C^{i}=1$ means $A_{i}$ is a pedestrian or cyclist. The trajectory has a historical trajectory and a real future trajectory. The sequence of observation trajectories $A_{i}$ in the $T_{obs}$ historical time step is $X=\left\{p_{i}^{t}\in R^{2}|t=1,2,\ldots ,T_{obs}\right\}$, where $p_{i}^{t}=\left(x_{i}^{t},y_{i}^{t}\right)$ represents the two-dimensional coordinates of the participant $A_{i}$ at time *t*. The purpose of this study is to generate a sequence of future $T_{pred}$ predicted trajectory from the time $T_{obs}$: $Y=\left\{{\hat{p}}_{i}^{t}\in R^{2}|t=T_{obs}+1,T_{obs}+2,\ldots ,T_{obs}+T_{pred}\right\}$, where ${\hat{p}}_{i}^{t}=\left({\hat{x}}_{i}^{t},{\hat{y}}_{i}^{t}\right)$.

Figure 1 shows the input and output of the trajectory prediction task (the figure is taken from the traffic scenario named USA_Roundabout_FT in the INTERACTION dataset). Figure 1a is the observation trajectory of multiple participants in 1 s history. Figure 1b is the trajectory of several traffic participants in the next 3 s. Among them, the blue and purple dots represent the current location of the vehicle and pedestrian/cyclist, respectively; the solid red line represents the historical trajectory, and the solid orange line represents the predicted trajectory.

### 3.2. Overall Network Framework

In real traffic scenarios, there are complex social interactions between pedestrians, vehicles, and other participants, which are difficult to model. We propose an STIA-TPNet to address this problem, which mainly consists of four parts shown in Figure 2.

Figure 2a is the trajectory feature extraction module, which encodes the historical trajectory and the real future trajectory, respectively, to capture the motion features. Figure 2b is the spatial–temporal interaction attention module, including two branches of temporal flow and space flow. Temporal interaction attention is used to capture the influence of historical trajectories at different moments on the motion of target participants, and spatial interaction attention is used to capture the social interaction of neighboring participants on the target. Fusing the two branches can obtain a spatial–temporal interaction attention representation. Figure 2c is the hidden variable generator, which uses conditional variational autoencoder sampling to obtain Gaussian hidden variables, which helps to generate multimodal trajectory prediction results. Figure 2d is the decoder, which is used to generate future trajectories.

### 3.3. Trajectory Feature Extraction

The trajectory feature extraction module takes the historical trajectory information of pedestrians, vehicles and other participants as input, and completes the extraction of key information by using embedding layers and LSTM networks.

Firstly, the coordinates $\left(x_{i}^{t},y_{i}^{t}\right)$ of the *i*th traffic participant within the historical time step are embedded into a high-dimensional vector $e_{i}^{t}$ using the embedding layer f(·),
(1)$$e_{i}^{t}=f\left(x_{i}^{t},y_{i}^{t};W_{emb}\right),$$
where, $x_{i}^{t}$ and $y_{i}^{t}$ represent the x and y coordinates of the traffic participant $A_{i}$ at time *t*; f(·) represents the embedding layer composed of linear layer and ReLU activation functions; and $W_{emb}$ represents the weight matrix of the embedding layer.

Then, the embedded high-dimensional feature vector $e_{i}^{t}$ is used as the input of the LSTM network to capture the motion features of the traffic participant.
(2)$$m_{i}^{t}=LSTM\left(m_{i}^{t-1},e_{i}^{t};W_{m}\right),$$
where, $m_{i}^{t}$ represents the hidden state vector of the LSTM network at time *t*, $W_{m}$ is the weight matrix of the LSTM network. To reduce complexity, these parameters are shared among all traffic participants in the scene.

The LSTM network is used to encode the real future trajectory sequence $Y_{R}=\left\{p_{i}^{t}\in R^{2}|t=T_{obs}+1,T_{obs}+2,\ldots ,T_{obs}+T_{pred}\right\}$ at the time steps $T_{obs}+1$ to $T_{obs}+T_{pred}$. The hidden state vector corresponding to the real future trajectory can be obtained.
(3)$$f_{i}^{t}=LSTM\left(f_{i}^{t-1},Y_{R};W_{f}\right),$$
where, $Y_{R}$ represents the real future trajectory sequence, $f_{i}^{t}$ represents the real future trajectory feature vector of the traffic participant $A_{i}$ at time *t*, and $W_{f}$ represents the weight matrix of the LSTM network.

### 3.4. Spatial-Temporal Interaction Attention Module

The trajectory feature extraction module only extracts the independent motion features of each traffic participant in the scene, but cannot capture the interaction relationship between traffic participants. We propose the Spatial–Temporal Interaction Attention Module (STIA Module) to extract the interaction relationships between traffic participants, including temporal interaction attention, spatial interaction attention, and spatio-temporal attention fusion. Figure 3 shows the diagram of the STIA Module.

**(1)** 
**Temporal interaction attention**


Pedestrians, vehicles and other traffic participants often have a fixed movement mode. Information for the previous time steps can help reason about the motion position of the next time step. Therefore, temporal context information is very important for trajectory prediction tasks, making the network pay different attentions to different parts of historical trajectories.

Temporal interaction attention is a temporal attention mechanism used to capture the movement pattern of each traffic participant in the scene; and by adaptive allocation of attention weights, it can learn the importance of historical trajectories at different moments to the future behaviors. This study uses the Self-Attention mechanism to model time dependence. Figure 4 shows the structure of temporal interaction attention.

The trajectory feature extraction module uses the LSTM network to encode the observation trajectory of each traffic participant at the historical time step $T_{obs}$. And the time context representation of the hidden state vector $m_{i}^{t}$ in the $T_{obs}$ time step can be obtained.
(4)$$M_{i}^{t}=\left[m_{i}^{1},m_{i}^{2},\ldots ,m_{i}^{T_{obs}}\right]\in R^{T_{obs}\times d},$$
where, $m_{i}^{t}$ is the output of Formula (2), which represents the hidden state vector of the *i*th traffic participant at time *t*, reflecting the historical trajectory characteristics of an traffic participant. $T_{obs}$ is the historical time step, and *d* is the dimension of each hidden state. $M_{i}^{t}$ represents the time context vector.

Taking the temporal context vector as input and processing it by using the self-attention mechanism, the attention weight coefficient $\alpha_{i}^{t}$ can be obtained, so as to adaptively learn important information from the time domain. By using the combination of Tanh and Softmax functions, the output value can be mapped to the normalized range of [0,1],
(5)$$\alpha_{i}^{t}=softmax\left(tan\left(W_{\alpha }M_{i}^{t}\right)\right)\in R^{1\times T_{obs}},$$
where, $W_{\alpha }$ represents a weight matrix with parameters shared by all traffic participants. softmax(·) represents the normalized exponential function, which ensures that the sum of all calculated weights is 1. tan(·) represents the Tanh activation function. $\alpha_{i}^{t}$ indicates the attention weight coefficient.

Finally, the attention weight coefficient and the time context vector are weighted summed, to obtain the vector representation of time attention,
(6)$$tm_{i}^{t}=\alpha_{i}^{t}M_{i}^{t}\in R^{1\times d},$$
where, $tm_{i}^{t}$ represents the temporal attention vector.

**(2)** 
**Spatial interaction attention**


In real traffic scenarios, vehicles and pedestrians interact with surrounding traffic participants. Therefore, effective modeling of the spatial interaction information of the target traffic participant and other participants is the key to predicting the future motion. Hence, we introduce a spatial interaction attention mechanism to capture spatial interaction information between all traffic participants at each time step.

Since the participants number in recent traffic scene dynamically changes, the self-attention mechanism in the time domain cannot be used to obtain the weight information of the spatial domain. Inspired by the mechanism of the graph attention network GAT [33], we abstract the traffic participants in the scene into graph nodes, and abstract the social interaction between traffic participants into graph edges. Figure 5 shows the structure diagram of spatial interaction attention. (a) shows the spatial interaction between several traffic participants, and the value next to the solid black line indicates the weight of the edge; (b) shows the calculation process of the attention weight coefficient; (c) shows the process of node feature fusion. Though the target may be affected by other neighboring traffic participants, just a part neighboring traffic participants affect the movement of the target; the spatial interaction attention method is constructed to adaptively assign weights to traffic participants in the scene, which indicates the degree of influence on the target movement.

In a graph with *N* nodes, the input to the graph attention layer is the initial feature of the node $m_{1:N}^{t}=\left\{m_{1}^{t},m_{2}^{t},\ldots ,m_{N}^{t}\right\},m_{i}^{t}\in R^{T_{obs}\times F}$, where $T_{obs}$ represents historical time and *F* represents the feature dimension of each node. After updating the node information, the output of the graph attention layer is $sm_{1:N}^{t}=\left\{sm_{1}^{t},sm_{2}^{t},\ldots ,sm_{N}^{t}\right\},sm_{i}^{t}\in R^{T_{obs}\times F^{\prime }}$, where $F^{\prime }$ represents the dimension of the output.

The motion state vector $m_{i}^{t}$ of the traffic participant in the historical observation time step is used as node information, and enter into the graph attention layer to obtain the interaction relationship between nodes. In order to learn the attention coefficient between node pairs (i,j), it is necessary to consider the influence of both nodes at the same time, and the calculation is
(7)$$e_{ij}^{t}=\alpha \left(Wm_{i}^{t},Wm_{j}^{t}\right)={\vec{\alpha }}^{T}[Wm_{i}^{t}||Wm_{j}^{t}],$$
where, α(·) represents a single-layer feed-forward neural network, ${\vec{\alpha }}^{T}$ represents the weight vector of a single-layer feed-forward neural network, || represents the concatenation operation, *W* represents the shared weight matrix between nodes, and $e_{ij}^{t}$ represents the attention value between node *i* and node *j*.

The loss of node *i* information may result in the attention weight between (i,j) not actually taking into account the information of node *i*. In order to avoid this problem, the LeakyReLU nonlinear activation function is used here. The attention coefficient obtained after the activation function operation is
(8)$$e_{ij}^{t}=LeakyReLU({\vec{\alpha }}^{T}[Wm_{i}^{t}||Wm_{j}^{t}]),$$
where, LeakyReLU(·) represents the LeakyReLU activation function, and the LeakyReLU activation function is
(9)$$LeakyReLU\left(x\right)=\left\{\begin{array}{ccc} x, & \; & x>0,\\ \alpha x, & \; & x\leq 0,\alpha =0.2, \end{array}\right.$$

After obtaining the attention coefficient between node *i* and its neighboring nodes, the Softmax activation function is used to regularize all attention coefficients. Figure 5b shows the process, and the calculation formula is
(10)$$\begin{array}{cc} \alpha_{ij}^{t} & =softmax(e_{ij}^{t})\\ \; & =\frac{exp(LeakyReLU({\vec{\alpha }}^{T}[Wm_{i}^{t}||Wm_{j}^{t}]))}{\sum_{k\in N_{i}}exp(LeakyReLU({\vec{\alpha }}^{T}[Wm_{i}^{t}||Wm_{k}^{t}]))}, \end{array}$$
where, softmax(·) is the normalized exponential function, exp(·) is an exponential function, $N_{i}$ represents the set of traffic participants adjacent to node *i*, and *k* represents the index of nodes in the set $N_{i}$.

Finally, the normalized attention coefficient is weighted with the original motion state vector, to obtain the updated feature vector representation of node *i*, which is the output result of the attention layer of the single-layer graph
(11)$$sm_{i}^{t}=\sigma \left(\sum_{j\in N_{i}}\alpha_{ij}^{t}Wm_{j}^{t}\right),$$
where, σ(·) represents the Sigmoid nonlinear activation function, and $sm_{i}^{t}$ represents the state vector that aggregates the trajectory features of neighboring traffic participants, reflecting the influence of the position of adjacent participants on the future motion of the target traffic participant.

In order to make the training learning process of attention more stable and prevent overfitting, the single-layer graph attention layer is expanded to multi-headed attention; and the transformation of Formula (11) is performed by introducing *K* independent attention mechanisms, and then the final spatial attention vector representation is obtained:(12)$$sm_{i}^{t,k}=\sigma \left(\frac{1}{K}\sum_{k=1}^{K}\sum_{j\in N_{i}}\alpha_{ij}^{t,k}W^{k}m_{j}^{t}\right),$$
where, *K* is the number of heads, *k* represents the *k*th head, $\alpha_{ij}^{t,k}$ is the normalization coefficient calculated by the *k*th attention mechanism, and $W^{k}$ represents the corresponding weight matrix.

Figure 5c shows the node feature fusion process of multi-head attention when K=3. Among them, the straight lines of different colors and line types are used to distinguish the independent attention calculation process of different heads; and it can be seen that the output result of spatial interaction attention can be obtained by connecting the features of each head on average.

**(3)** 
**Temporal and spatial attention fusion**


Temporal interaction attention can explain the influence of historical trajectory on the future trajectory, but cannot well represent the spatial interaction between traffic participants. Spatial interaction attention acts on the spatial dimension and ranks neighboring participants according to their influence on the target traffic participant, which can effectively captures the spatial interaction information; yet, the motion of a single traffic participant cannot be well modeled, so coupling the temporal-spatial interaction information is very important to achieve accurate trajectory prediction.

In order to adaptively select more important information in both the spatial domain and the time domain, the vector $tm_{i}^{t}$ obtained by temporal interaction attention is fused with the vector $sm_{i}^{t}$ obtained by spatial interaction attention; and finally the temporal–spatial attention vector $st_{i}^{t}$ can reflect temporal–spatial interaction.
(13)$$st_{i}^{t}=sm_{i}^{t}\left[\begin{array}{c} tm_{1}^{t}\\ tm_{2}^{t}\\ \ldots \\ tm_{N}^{t} \end{array}\right].$$

### 3.5. Hidden Variable Generator

The motion of traffic participants have multimodal features, and there are a lot of uncertainties in the predicted future trajectories. Most of the existing trajectory prediction methods use hidden variables to generate multimodal prediction results. The hidden variables often ignore information about social interactions, and have insufficient information about contextual relationship. In order to make the hidden variables reflect the influence of social interaction, the Conditional Variational AutoEncoder (CVAE) [34] is used in this model to learn the parameters of the sampling distribution from historical trajectories and spatio-temporal interaction information.

The conditional variational autoencoder consists of three parts: the priori network $P_{\theta }\left(Z|X\right)$, the recognition network $Q_{\Phi }\left(Z|X,Y\right)$ and the generation network $P_{\Psi }\left(Y|X,Z\right)$; and they are implemented through a fully connected layer.

In the training phase, the temporal–spatial attention vector $st_{i}^{t}$ and the real future trajectory vector $f_{i}^{t}$ of the traffic participant are connected in series and jointly input into the recognition network $Q_{\Phi }\left(Z|X,Y\right)$, to predict mean $\mu_{Z}^{q}$ and covariance $\sigma_{Z}^{q}$,
(14)$$\left(\mu_{Z}^{q},\sigma_{Z}^{q}\right)=Q_{\Phi }^{MLP}\left(st_{i}^{t}⊕f_{i}^{t}\right),$$
where, $Q_{\phi }^{MLP}\left(\cdot \right)$ represents the identification network implemented with MLP. In the testing phase, since the real future trajectory sequence $Y_{R}$ is unknowable, only the spatio-temporal attention vector $st_{i}^{t}$ is used as the input of the priori network $P_{\theta }\left(Z|X\right)$, obtaining the mean $\mu_{Z}^{p}$ and the covariance $\sigma_{Z}^{p}$.
(15)$$\left(\mu_{Z}^{p},\sigma_{Z}^{p}\right)=P_{\theta }^{MLP}\left(st_{i}^{t}\right),$$
where, $P_{\theta }^{MLP(\cdot )}$ represents a priori network implemented with MLP. Through the training and learning of the network, the KL divergence loss between $N(\mu_{Z}^{p},\sigma_{Z}^{p})$ and $N(\mu_{Z}^{q},\sigma_{Z}^{q})$ is continuously optimized, and finally the hidden variable *Z* is sampled.

### 3.6. Decoder

In order to enhance the anti-interference and noise reduction ability of the decoder, the spatio-temporal attention $st_{i}^{t}$ and the hidden variable *Z* concat together, and input into the GRU network [35].
(16)$$stz_{i}^{t}=st_{i}^{t}⊕Z,$$
(17)$$d_{i}^{t}=GRU\left(d_{i}^{t-1},stz_{i}^{t};W_{dec}\right),$$
where, $d_{i}^{t}$ represents the hidden state vector of the GRU network at time *t*, and $W_{dec}$ represents the weight of the GRU network. The hidden state vector output by the GRU network is input into the linear layer, and the future motion position of each traffic participant can be obtained.
(18)$$\left({\hat{x}}_{i}^{t},{\hat{y}}_{i}^{t}\right)=\delta \left(d_{i}^{t};W_{\delta }\right),$$
where, ${\hat{x}}_{i}^{t}$ and ${\hat{y}}_{i}^{t}$ represent the horizontal and ordinate coordinates of the future position predicted by the traffic participant $A_{i}$ at time *t*, and δ(·) represents the linear layer, $W_{\delta }$ is the weight of the linear layer.

## 4. Experimental Results and Analysis

### 4.1. Dataset

**(1) INTERACTION dataset**. The INTERACTION dataset [18] was built by the MSC Lab at the University of California, Berkeley, which contains many internationalized, diverse, complex and critical driving scenarios and vehicle motions. The motion data were collected from different countries and continents. The dataset collects motion data from four different traffic scenarios, including traffic circles, unsignalized intersections, merging and lane changing, and signalized intersections, involving a total of eleven different collection shot locations. In the traffic circle scenario, 10,479 motion trajectories from five different locations were recorded, with a duration of about 365 min; in the unsignalized intersection scenarios with three different locations, 14,867 trajectories were collected over a period of about 433 min; in the merge and lane change scenario, the data were taken from two different locations, which contained a total of 10,933 trajectories, with a duration of approximately 133 min; in the signalized intersection scenario, only one location was selected and it provided 3775 trajectories over approximately 60 minutes.

**(2) UTP-dataset**. The UAV-captured Trajectory Prediction Dataset (UTP-dataset) is a captured dataset based on UAV aerial photography in real urban scenarios, with 97.34 min videos, 2512 pedestrians and 2778 vehicles. In order to detect moving intelligences in videos, pedestrians, cyclists, and vehicles are detected and localized, and the position and orientation of each traffic participant is estimated using oriented bounding boxes. Subsequently, the Kalman filter and the Hungarian algorithm are used to achieve multi-target tracking. In order to facilitate the use of data for the model, the UTP dataset was divided into training, validation, and testing sets in order of 70%, 10%, and 20%.

### 4.2. Evaluation Criteria and Experimental Settings

The Average Displacement Error (ADE) and Final Displacement Error (FDE) are selected to evaluate the performance of the trajectory prediction performance; the smaller values of these two parameters means more accurate prediction performance of future trajectories.

(1) Average Displacement Error (ADE): ADE is used to calculate the average Euclidean distance between the real trajectory and the predicted trajectory in the predicted time step, reflecting the average prediction performance.
(19)$$ADE=\frac{\sum_{i=1}^{N}\sum_{t=1}^{T_{pred}}\left|\right|{\hat{p}}_{i}^{T_{obs}+t}-p_{i}^{T_{obs}+t}{\left|\right|}_{2}}{N\times T_{pred}},$$
where, ${\hat{p}}_{i}^{T_{obs}+t}$ and ${\hat{p}}_{i}^{T_{obs}+t}$ represent the predicted trajectory and real future trajectory of the traffic participant $A_{i}$ at the $T_{obs}+t$ time, *N* represents the number of traffic participants in the scene, and the $T_{obs}$ and $T_{pred}$ represent the observation time and prediction time, respectively.

(2) Final Displacement Error (FDE): FDE is used to calculate the average Euclidean distance between the real trajectory and the predicted trajectory at the final position, reflecting the final prediction accuracy of the model,
(20)$$FDE=\frac{\sum_{i=1}^{N}\left|\right|{\hat{p}}_{i}^{T_{obs}+T_{pred}}-p_{i}^{T_{obs}+T_{pred}}{\left|\right|}_{2}}{N},$$
where, ${\hat{p}}_{i}^{T_{obs}+T_{pred}}$ and $p_{i}^{T_{obs}+T_{pred}}$, respectively, represent the predicted trajectory and real future trajectory of the traffic participant $A_{i}$ at the final moment of prediction $T_{obs}+T_{pred}$, *N* represents the number of traffic participants in the scene, and $T_{obs}$ and $T_{pred}$ represent the observation time and prediction duration, respectively.

### 4.3. Experimental Settings

The proposed STIA-TPNet method is based on the Pytorch. The hardware platform is a server equipped with an Intel Core i9-9900X CPU and three NVIDIA TITAN XP graphics cards. The operating system is Ubuntu 16.04 LTS, and the graphics driver version is CUDA 9.0. The software development platform is Pycharm2022.

The UTP dataset and the INTERACTION dataset are divided into training sets, validation sets and test sets, according to the proportions of 70%, 10% and 20%. For UTP datasets, we set parameters $T_{obs}=30$ and $T_{pred}=30$; and for INTERACTION datasets, we set parameters $T_{obs}=10$ and $T_{pred}=30$.

For neural network structure and parameters, the input dimension of Equation (Equation 1) is 2, the output dimension is 32. The dimension of the LSTM encoder in Equation (Equation 2) is 128. The bidirectional LSTM encoder with dimension 32 is used in Equation (Equation 3). The dimension of the GRU decoder is 256. Spatial interaction attention contains two graph attention layers; for the first layer, the input dimension of the first layer is 128, the output dimension is 64, and the number of attention heads K=4; for the second layer, the input dimension is 256, the output dimension is 128, and the number of attention heads K=1. In the hyperparameter settings, the size of batch size is set to 256, which is a large setting according to the GPU memory; the size of the number of iterations is set to 150, and the learning rate is 0.001. With these settings, the loss value decreases fast and stable.

### 4.4. Trajectory Prediction Results on the UTP Dataset

In order to verify the effectiveness of the proposed method, some trajectory prediction methods are used for comparison, including Vanilla LSTM [36], Social LSTM [5], Social GAT [8], STGAT [37], Social STGCNN [11], DisDis [38], GSTA [25], Cae-gan [26], Social Implicit [39] and Trajectron++ [13].

Vanilla LSTM [36]: The basic LSTM encoder–decoder model that uses an LSTM network to encode the motion of a single target, and uses the LSTM decoder to obtain the predicted future trajectory of the participant in the decoding stage, without considering the social interaction between traffic participants.

Social LSTM [5]: The trajectory prediction model based on social LSTM uses the social interaction pooling layer to combine and share the state information of adjacent sequences, and effectively models the interaction effect between participants.

Social GAN [8]: The generative adversarial model that combines LSTM and GAN, and introduces distance-based pooling modules to obtain participant interaction representations. Among them, the generator generates multiple prediction trajectories, and the discriminator distinguishes the true and false trajectories.

STGAT [37]: The LSTM encoder–decoder model based on graph attention mechanism uses graph attention network to assign different weights to participants, and effectively models the local interaction and global interaction between participants.

Social STGCNN [11]: The trajectory prediction model based on a social spatio-temporal graph convolutional neural network firstly extract features through spatio-temporal convolution operations and then decodes the spatio-temporal features to obtain the prediction trajectory.

DisDis [38]: The trajectory prediction model based on the distribution discrimination method can predict the potential distribution and motion patterns of traffic participant personalization, through self-supervised contrast learning and discrete optimization of consistency constraints.

Social Implicit [39]: The socially implicit model includes social regions, social units, and social loss, and introduces implicit maximum likelihood estimation for training the model, which can generate trajectory prediction results close to the real trajectory.

Trajectron++ [13]: The trajectory prediction model based on graph structure and LSTM network uses LSTM encoding and extracting spatio-temporal information, and then uses a conditional variational autoencoder to generate multiple reasonable and feasible future trajectories.

GSTA [25]: The spatio-temporal graph attention (GSTA) network combines GAN and LSTM to design the spatial attention structure and feature an updating mechanism in the spatial domain, and temporal attention and feature selection module in the temporal domain. Experimental results show the GSTA improved the pedestrian trajectory prediction accuracy.

Cae-gan [26]: The mixed Conditional Auto Encoder Generative Adversarial Network (CAE-GAN) model based on the multi-loss function is proposed. The GAN in the CAE-GAN is used to extract features of the generated trajectories and generate trajectories that are close to the real trajectories.

The proposed STIA-TPNet model is compared with these methods on the UTP dataset, as shown in Table 1, where the best results are highlighted in bold. The proposed trajectory prediction method obtains an ADE distance of 5.56 pixels and an FDE distance of 10.21 pixels. Compared with other methods, our method reduces the average displacement error by 4.5% and the final displacement error by 1.0%. There are several reasons that contribute to this achievement. The temporal interaction attention can learn the importance of historical trajectories at different moments to the future behaviors. The spatial interaction attention can abstract the traffic participants and social interaction into graph nodes and edges. Coupling the temporal and spatial interaction attentions can adaptively model the temporal–spatial information and achieve accurate trajectory prediction.

In order to evaluate the influence of the spatio-temporal interaction attention module (STIA Module), ablation experiments are performed, as shown in Table 2. Baseline is a model without adding a temporal–spatial interaction attention module; STIA-TPNet is a model that introduces the temporal–spatial interaction attention module on the basis of the baseline.

By analyzing the results of Table 2, it can be seen that the spatio-temporal interaction attention module has 0.28% better ADE and 0.1% better FDE. This result means that the STIA-TPNet can effectively capture the interaction between time domain and space domain. Therefore, the STIA-TPNet method achieves the optimal prediction error and can generate multiple prediction trajectories that conform to physical and social constraints.

In order to better analyze the effect of our method, we visualize some scenarios in the UTP dataset, such as Figure 6. As shown, the solid red line in the figure represents the observed 3 s historical trajectory, the blue-dashed line represents the real future 3 s trajectory, and the solid orange line represents the predicted 20 future trajectories. In Figure 6a, vehicle 801 and vehicle 802 turn left and right at the intersection, respectively, in order to avoid interfering with the movement of the other party; vehicle 801 turns slightly along the left side of the road, and vehicle 802 turns along the right side of the road. In Figure 6b, vehicle 821 and vehicle 865 turn right at the same time, and the predicted trajectory can effectively avoid collisions. Therefore, the proposed STIA-TPNet trajectory prediction network can generate reasonable future trajectories in these complex traffic scenarios. The main reason is that our method can effectively capture the complex interaction information between traffic participants.

### 4.5. Trajectory Prediction Results on the Interaction Dataset

STIA-TPNet trains the resulting loss function curve on the INTERACTION dataset, shown in Figure 7. As the number of iterations gradually increases, the curve shows a tendency to decline and converge. This curve shows that our method can converge steadily with more than 50,000 training iterations.

Table 3 is the comparison results of trajectory predictions on the INTERACTION dataset. Smaller values indicate higher accuracy of trajectory prediction, and the optimal method is indicated in bold. It can be seen that the average displacement error and final displacement error calculated by the proposed algorithm are 0.45 m and 0.99 m, respectively. The spatio-temporal interaction attention module is used to effectively extract the interaction information between traffic participants, which improves the prediction accuracy of the model. Compared to other methods, the STIA-TPNet method reduces ADE by 13.5% and FDE by 15.4%. Based on trajectory feature extraction, the STIA Module can extract the interaction relationships between traffic participants. The main reason is that coupling the temporal and spatial interaction attentions can adaptively model the temporal–spatial information and achieve accurate trajectory prediction. Hence, our method can achieve the state-of-the-art results on both the UTP dataset and the INTERACTION dataset.

The interval of 1 s to 3 s in the prediction time domain is divided at 0.5 s intervals, and the STIA-TPNet algorithm is experimented on the INTERACTION dataset. Figure 8 shows the variation curves of the average displacement error (ADE) and final displacement error (FDE) of the STIA-TPNet method with the predicted time domain; the solid orange and green lines correspond to ADE and FDE, respectively. The prediction process is often under complex and changeable traffic scenarios. The trajectory prediction is sensitive to prediction time, and a longer prediction time often means a larger prediction error. As the prediction time increases, the average displacement error and the final displacement error gradually increase. Hence, in the trajectory prediction task, the prediction time needs to be considered according to different applications.

Figure 9 shows the trajectory prediction results of some traffic scenarios in the INTERACTION dataset; the solid red line is the observation trajectory; the blue dashed line is the real future trajectory; the solid orange line is the predicted multiple future trajectories; the blue dot represents the vehicle; and the purple dot represents the pedestrian or cyclist. Figure 9a corresponds to the scene USA_Roundabout_SR, where pedestrians 2 and 4 are about to enter and leave the roundabout, so the predicted trajectory travels along the inside of the roundabout. Figure 9b is the scenario CHN_Roundabout_LN, where vehicle 3 exits the roundabout from the intersection, and the prediction results are in line with the movement trend. Figure 9c corresponds to the scene CHN_Merging_ZS, where vehicles 1, 4, 8, and 16 drive normally along the lane lines. Figure 9d is the scenario DEU_Merging_MT, where the road changes from a two-lane to a single lane, so vehicles 2 and 5 drive on the right side of the single lane. The proposed method can predict trajectories measured by time. Considering vehicle speed, faster vehicles have longer predicted trajectories. Although there is a certain deviation between the predicted trajectory and the actual trajectory, the overall trend of the predicted trajectory is close to the real situation; these prove the effectiveness of our trajectory prediction network based on spatio-temporal interaction attention, which can generate accurate and reliable predicted trajectories. The main reason is that the spatio-temporal interaction attention can generate the most possible trajectory from a batch of trajectories.

## 5. Discussion

The proposed method can generate accurate and reliable predicted trajectories. Some reasons may result in failures, including traffic light change, breaking traffic regulations, environmental constraint change. These reasons can not reflect on history trajectories, and may result in wrong predictions.

The proposed trajectory prediction can be applied in drone-based applications for intelligent driving, traffic optimization, intelligent monitoring, etc. In this study, all calculation is processed with our GPU server and the calculation time is 5.6 fps. Though the consuming time problem is not mainly considered in this study, we argue that cloud and edge calculation can also be used to promote the calculation ability in real applications. For example, the capture videos can be preprocessed on the drone and then sent to the server or cloud for further calculation.

## 6. Conclusions and Future Work

For traffic scenarios, pedestrians, vehicles and other traffic participants have social interactions of surrounding traffic participants in both time and spatial dimensions. In order to simulate the spatial–temporal interactions and capture the spatio-temporal dependence, we propose the Spatial–Temporal Interaction Attention-based Trajectory Prediction Network (STIA-TPNet).

The STIA-TPNet includes a temporal flow and a spatial flow. In the temporal flow, temporal interaction attention is used to analyze the importance of the historical trajectories to the future movement of the target participant; spatial interaction attention is to capture the interaction effect of neighboring traffic participants. Fusing the spatial–temporal interaction information can adaptively select the important information in the both time and spatial domains. Experiments on the two datasets show that the proposed method significantly improves the accuracy of trajectory prediction. In this study, the dataset size are still limited and the calculation ability in real applications is not fully considered. In the future, we aim to further improve the generalization ability with more training samples from different scenarios, and further re-implement our method in real applications.

## Figures and Tables

**Figure 1 sensors-23-07830-f001:**
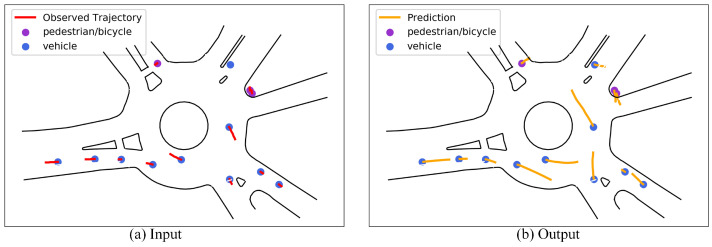
Schematic diagram of the input and output of the trajectory prediction task.

**Figure 2 sensors-23-07830-f002:**
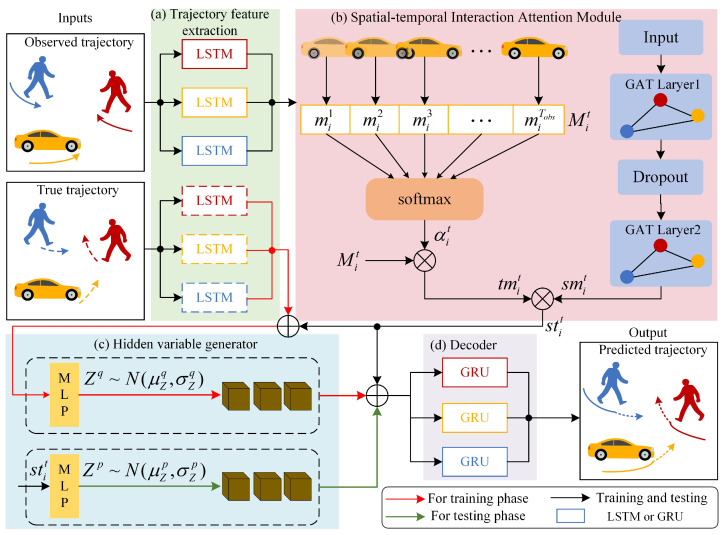
Traffic participant trajectory prediction network based on spatial–temporal interaction attention.

**Figure 3 sensors-23-07830-f003:**
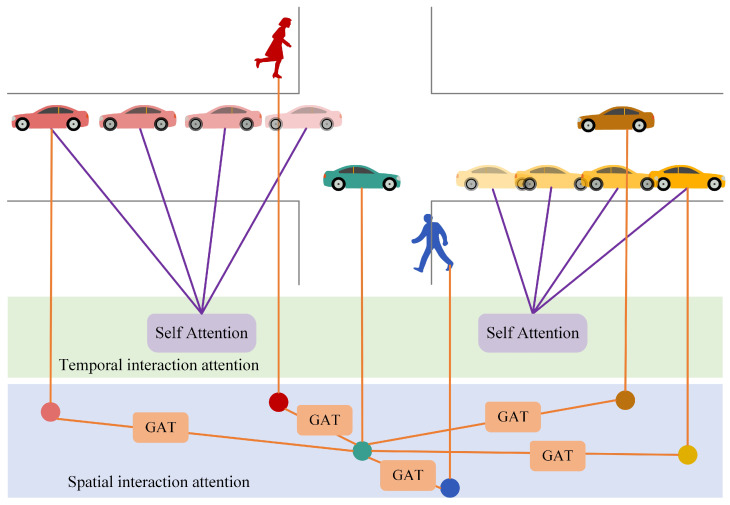
Schematic diagram of spatial–temporal interaction attention module.

**Figure 4 sensors-23-07830-f004:**
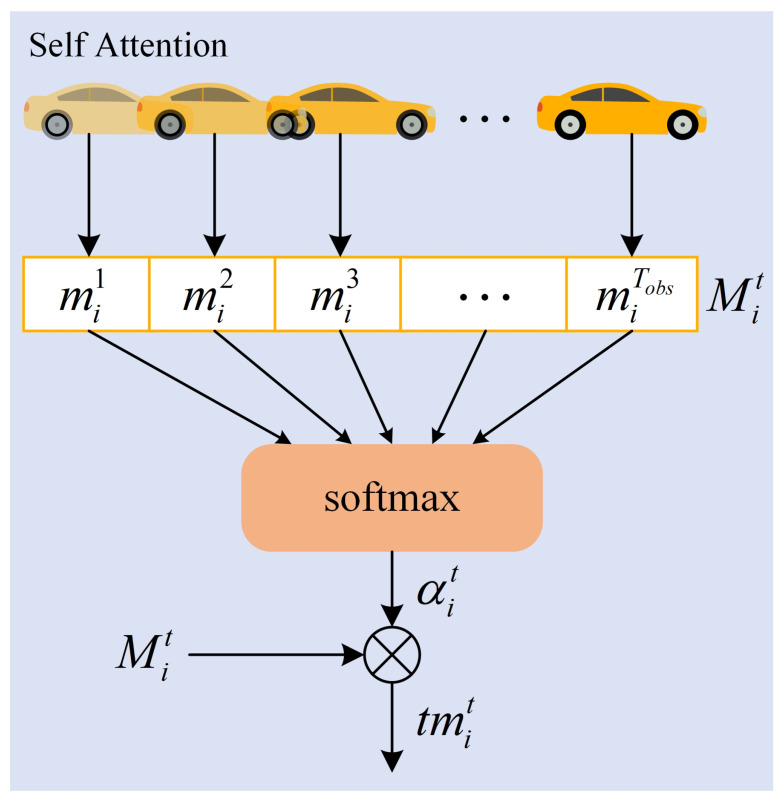
Temporal interaction attention structure diagram.

**Figure 5 sensors-23-07830-f005:**
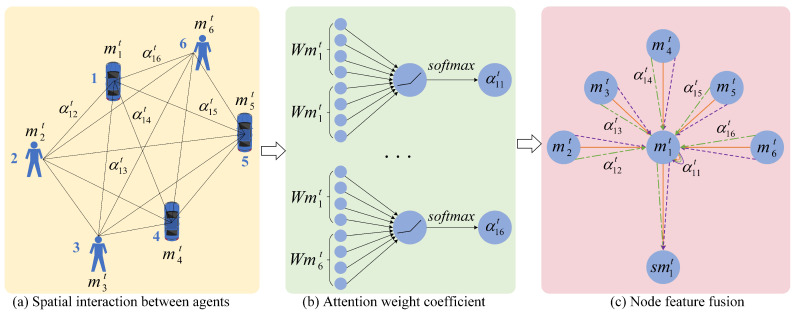
Spatial interaction attention structure diagram.

**Figure 6 sensors-23-07830-f006:**
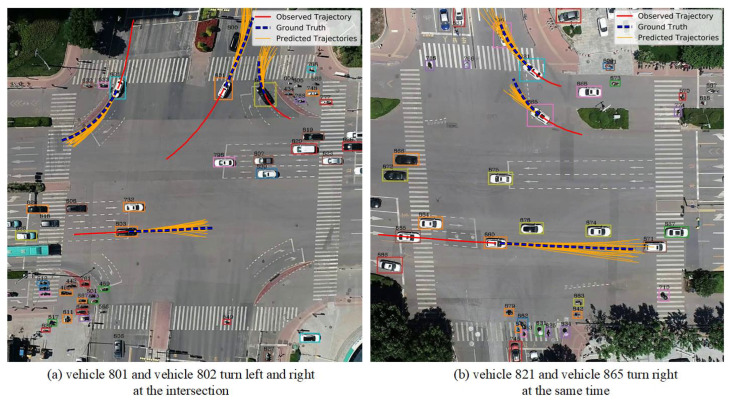
Visualization results of trajectory prediction on UTP dataset.

**Figure 7 sensors-23-07830-f007:**
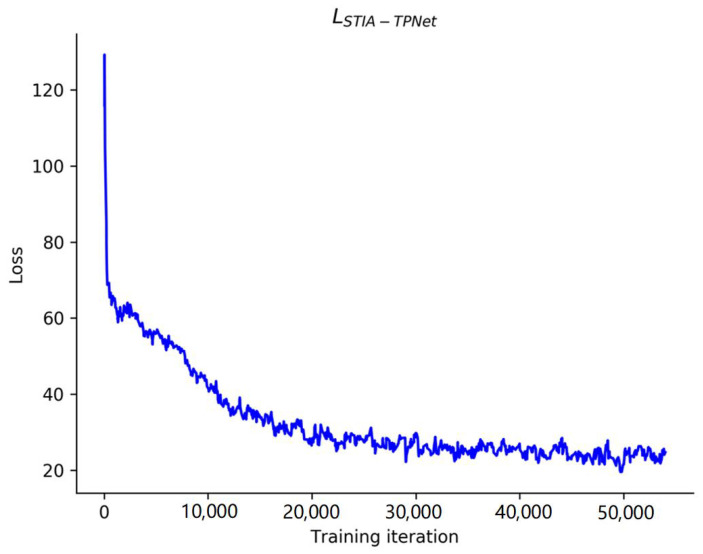
Loss function curve of the STIA-TPNet model on the INTERACTION dataset.

**Figure 8 sensors-23-07830-f008:**
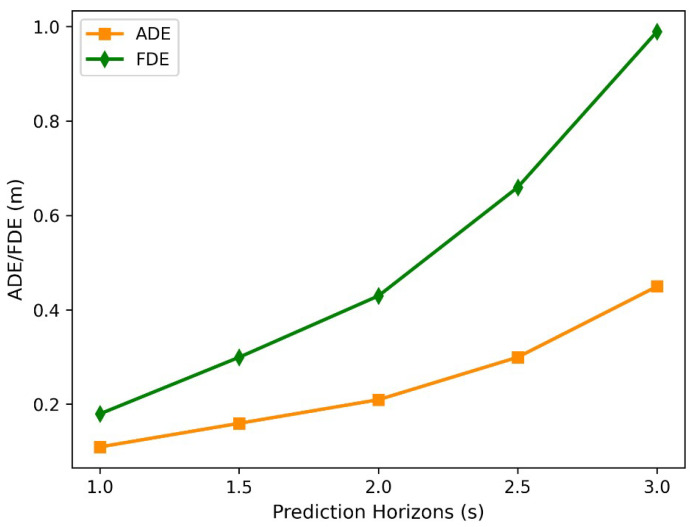
ADE and FDE of different prediction horizons on the INTERACTION dataset.

**Figure 9 sensors-23-07830-f009:**
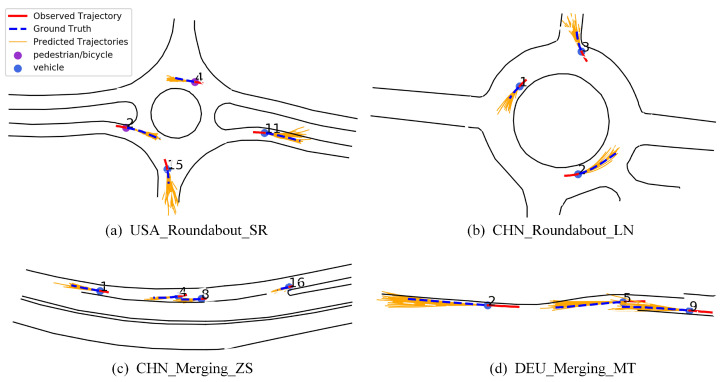
Visualization results of trajectory prediction on INTERACTION dataset.

**Table 1 sensors-23-07830-t001:** Comparison of ADE and FDE results of different methods on UTP dataset.

Method	ADE ↓	FDE ↓
Vanilla LSTM [36]	10.06	23.86
Social LSTM [5]	10.04	23.67
STGAT [37]	10.82	24.67
Social STGCNN [11]	14.29	29.51
DisDis [38]	5.82	13.89
Social Implicit [39]	11.22	27.86
GSTA [25]	10.67	21.12
Cae-gan [26]	9.56	18.23
Trajectron++ [13]	5.82	10.31
**STIA-TPNet (Ours)**	**5.56**	**10.21**

**Table 2 sensors-23-07830-t002:** Ablation experiment of STIA-TPNet on UTP dataset.

Method	STIA Module	ADE ↓	FDE ↓
Baseline		5.82	10.31
STIA-TPNet	✓	**5.56**	**10.21**

**Table 3 sensors-23-07830-t003:** Comparison of ADE and FDE results of different methods on INTERACTION dataset.

Method	ADE ↓	FDE ↓
Vanilla LSTM [36]	0.84	2.14
Social LSTM [5]	0.84	2.11
Social GAN [8]	0.83	1.97
STGAT [37]	0.99	2.38
Social STGCNN [11]	1.20	2.58
DisDis [38]	0.70	1.66
GSTA [25]	1.10	2.06
Cae-gan [26]	0.98	1.96
Social Implicit [39]	0.58	1.40
Trajectron++ [13]	0.52	1.17
**STIA-TPNet (Ours)**	**0.45**	**0.99**

## Data Availability

Not applicable.

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
