# Peer review of "Traffic Agents Trajectory Prediction Based on Spatial–Temporal Interaction Attention"

_sensors, 2023, doi:10.3390/s23187830_

Round 1
Reviewer 1 Report
· The language usage throughout this paper need to be improved, the author should do some proofreading on it. Give the article a mild language revision to get rid of few complex sentences that hinder readability and eradicate typo errors.
· Overall, the basic background is not introduced well, where the notations are not illustrated much clear. I recommend the authors to employ certain intuitive examples to elaborate the essential notations.
· Spell out each acronym the first time used in the body of the paper. Spell out acronyms in the Abstract.
· The abstract can be rewritten to be more meaningful. The authors should add more details about their final results in the abstract. Abstract should clarify what is exactly proposed (the technical contribution) and how the proposed approach is validated.
· What is the motivation of the proposed work? Research gaps, objectives of the proposed work should be clearly justified.
The authors should consider more recent research done in the field of their study (especially in the years 2022 and 2023 onwards).
· Introduction needs to explain the main contributions of the work more clearly.
· The novelty of this paper is not clear. The difference between present work and previous Works should be highlighted. In the references in the Introduction section, the authors cite some works. However, they have not indicated the advantage or disadvantage and their relations to this paper. It’s a little confusing.
· The author has mentioned the errors obtained by used techniques, it is suggested that the significance of errors listed, must be described in the comparison section.
· The major trends of the simulation should be noted using bullet points.
· Comparsion with recent study and methods would be appreciated.
· Introduction section can be extended to add the issues in the context of the existing work and how proposed algorithms/approach can be used to overcome this. Literature review techniques has to be strengthened by including the issues in the current system and how the author proposes to overcome the same.
· The paper does not explain clearly its advantages with respect to the literature: it is not clear what is the novelty and contributions of the proposed work: does it propose a new method? Or does the novelty only consists in the application?
· The advantage of the proposed method with respect to other methods in the literature should be clarified.
· The paper does not provide significant experimental details needed to correctly assess its contribution: What is the validation procedure used?
· Results need more explanations. Additional analysis is required at each experiment to show the its main purpose.
· Clarify the finding Error rate and accuracy in performance analysis section.
· Introduce the chart for given algorithm with description.
· Authors should add more details about the implementation of the code to perform the analysis and the library involved in this task.
· The results of analyzes were presented in a consistent and explicit form using graphs and tables, but selection of research for analysis raises objections.
· Experimental results are not clear. What are the parameters used in the proposed system and how their values are set? Also, how the parameter values can affect the proposed system? Sections like Experimentation have to be extended and improved thus providing a more convincing contribution to the paper.
· an error and statistical analysis of data should be performed.
· a comparison with state of art in the form of table should be added
correlate it with other current Technologies, such as: Blockchain, IoT (communications, networks, Cloud, …), in terms of latency I guess that this field is quite sensitive to the delays required to process data, which should call for new investigations around the tradeoff between learning cost and performance (e.g. Deep Learning is costly, yet attains good predictive scores… should we opt for weak learners over good features? Or complex learners over raw data? Or a mixture of both of them, e.g. learned features off-line + weak learners on-line? Should data be sent to the cloud? Be preprocessed at the edge?). This issue is also very trendy at the communications level.
must be improved
Author Response
First of all, we would like to thank the Reviewer for the comments and constructive suggestions. Based on these comments, we have made modifications in the revised manuscript. We hope that all concerns can be addressed in a satisfying way. Below the reviewers’ original text is included in Bold font. The answers are given in normal font. The changed parts are marked in red. The sequence numbers of figures and equations are corresponding to that in the revised manuscript.

Reviewer 2 Report
Traffic Agents Trajectory Prediction based on
Spatial-Temporal Interaction Attention
In this manuscript, in order to simulate the spatial-temporal interactions and capture the spatio-temporal dependence, the author proposes the Spatial-Temporal Interaction Attention based Trajectory Prediction Network (STIA-TPNet). The STIA-TPNet includes a temporal flow and a spatial flow. In the temporal flow, temporal interaction attention is used to analyze the importance of the historical trajectories to the future movement of the target agent; spatial interaction attention is to capture the interaction effect of neighboring agents. Fusing the spatial-temporal interaction information can adaptively select the important information in the both time and spatial domains.
The research is complete in structure and rich in content, and can also provide educational instructions for the society. Besides, there are sufficient analysis to explain the conclusions, which is easy for readers to understand.
To sum up, considering the whole quality of this manuscript and the standard of this journal, I suggest that it should be accepted.
Traffic Agents Trajectory Prediction based on
Spatial-Temporal Interaction Attention
In this manuscript, in order to simulate the spatial-temporal interactions and capture the spatio-temporal dependence, the author proposes the Spatial-Temporal Interaction Attention based Trajectory Prediction Network (STIA-TPNet). The STIA-TPNet includes a temporal flow and a spatial flow. In the temporal flow, temporal interaction attention is used to analyze the importance of the historical trajectories to the future movement of the target agent; spatial interaction attention is to capture the interaction effect of neighboring agents. Fusing the spatial-temporal interaction information can adaptively select the important information in the both time and spatial domains.
The research is complete in structure and rich in content, and can also provide educational instructions for the society. Besides, there are sufficient analysis to explain the conclusions, which is easy for readers to understand.
To sum up, considering the whole quality of this manuscript and the standard of this journal, I suggest that it should be accepted.
Author Response
We are honored to receive your agreement of our paper. We have also improved our paper according to your comments and other reviewers' comments.

Reviewer 3 Report
The information presented in this article is useful, research contribution is satisfactory. However, the following suggestions are recommended:
1- Your abstract does not highlight the specifics of your research or findings. Rewrite the Abstract section to be more meaningful. I suggest: problems, Aim, Methods, Results, and Conclusion.
2- The introduction section should contain a critical analysis of previous literature for what has been done, research gaps, and limitations, to justify the novelty of your work. It should be expanded to include a more detailed discussion of current problems.
3- In the results and discussion section, the author is encouraged to provide more in-depth discussions about each figure, and not just an interpretation of what is shown in the Figures. In addition, the result trends should be compared with the existing literature; this is to ensure that they offer expected outcomes.
4- Authors need to explain in detail what have done and what have not been done in other studies. What is the different within your study and other study?
5- What are the implications of the obtained results?
6- The Limitations of the proposed study need to be discussed before conclusion.
7- The authors should suggest possible areas of further research.
8- The authors should include the applications of the results obtained.
9- You need to review your English.
Minor editing of English language required.
Author Response

(The authors gave the same response as above.)

Reviewer 4 Report
The paper is a study on the traffic agents trajectory prediction based on spatial-temporal interaction attention and is considered a valuable and interesting study in related fields. The reviewer's opinions are as follows.
1. Abstract should be concisely and clearly described, including the background, purpose, method, result, and conclusion of the study.
2. In the description, ambiguous expressions should be avoided and quantitative numerical values or objective grounds should be presented.
3. It is necessary to describe existing efforts(papers) regarding the problems (not the simple description of the existing studies). The methods that solved the problems perceived in previous similar studies should be described in detail(academic excellence on this paper).
4. In the 'conclusion' part, it is necessary to describe the limitations of the study and additional studies required in the future. It is recommended to describe the interpretation of the research results in an easy-to-understand manner.
Thank you very much.
Ambiguity should be avoided in descriptions, and some terms used may be interpreted differently (e.g. agent, etc.). The expression style overall should be checked and minor editing of English language is required.
Author Response

(The authors gave the same response as above.)

Round 2
Reviewer 1 Report
Accept as it is.